# RefChess: Training-Free Contextual Search for Zero-Shot Referring Image Segmentation

**Shiyan Tong** [1]  **Jinxia Zhang** [1 2]  **Zhiyuan Wang** [3]  **Hao Tian** [4]  **Yingying Wang** [1]  **Kanjian Zhang** [1]  **Haikun Wei** [1]

## Abstract

Recent advances in zero-shot referring image segmentation (RIS), driven by foundation models such as SAM and CLIP, have improved cross-modal alignment between visual regions and natural language expressions. Nevertheless, selecting the correct segmentation proposal remains challenging, as existing methods typically score proposals independently and can be distracted by visually similar candidates that partially satisfy the expression. To address this limitation, we propose RefChess, a training-free contextual search framework for robust proposal selection. Instead of treating proposal selection as a single-step ranking problem, RefChess evaluates candidate masks under sampled distractor contexts and uses Monte-Carlo Tree Search as a budgeted mechanism to explore the combinatorial space of contextual perturbations. The search is guided by a stability-aware reward that integrates language decomposition, vision–language similarity, object-centric cues, and spatial guidance signals. Experiments on standard RIS benchmarks show that RefChess consistently improves robustness and referring segmentation performance without task-specific training. Code is available at https://github.com/Tongshiyan/RefChess.

## 1 Introduction

Referring Image Segmentation (RIS) aims to identify and segment a target object in an image according to a nat-

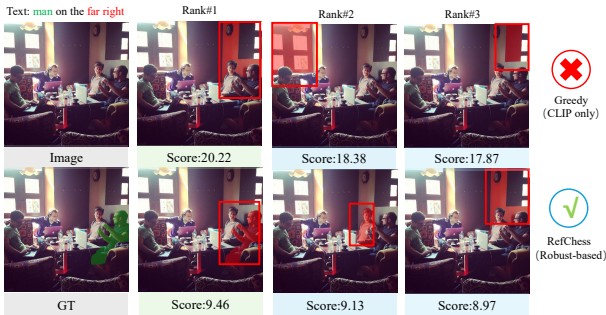

*Figure 1.* Top-3 proposals (with scores) for the same query. CLIP-only ranking selects regions inconsistent with the expression, while robust-based ranking selects the correct instance more reliably.

ural language expression. As a fundamental problem in vision–language understanding, RIS plays an important role in applications such as visual search, human–robot interaction, and assistive perception. Recently, the emergence of large-scale foundation models, most notably the Segment Anything Model (SAM) (Kirillov et al., 2023) and CLIP (Radford et al., 2021), has significantly advanced RIS by enabling segmentation without task-specific training or annotated data.

Recent advances in RIS have substantially improved region–text alignment, benefiting from the complementary strengths of foundation models such as SAM and CLIP. Building upon this paradigm, many works (Suo et al., 2023; Ni et al., 2024; Sun et al., 2024; Liu & Li, 2025) first generate instance-level mask proposals using SAM and then select the final prediction by ranking proposals according to their semantic similarity with the referring expression computed by CLIP-based models (Radford et al., 2021; Li et al., 2022). Although effective in many cases, this paradigm treats proposal evaluation as an independent scoring problem, overlooking interactions among competing candidates. In complex scenes, where multiple regions share similar visual appearance or satisfy partial linguistic constraints, such independent scoring often leads to unstable or incorrect selections, as shown in Fig. 1.

Beyond the formulation perspective, another key challenge in RIS lies in effectively extracting and organizing heteroge-

---

[1]Key Laboratory of Measurement and Control of CSE, Ministry of Education, School of Automation, Southeast University, Nanjing, China [2]Advanced Ocean Institute of Southeast University, Nantong, China [3]School of Computer Science & Engineering, University of Electronic Science and Technology of China, Chengdu, China [4]School of Automation, University of Electronic Science and Technology of China, Chengdu, China. Correspondence to: Jinxia Zhang <jinxiazhang@seu.edu.cn>.

*Proceedings of the 43rd International Conference on Machine Learning*, Seoul, South Korea. PMLR 306, 2026. Copyright 2026 by the author(s).

neous semantic cues from both language and vision. Referring expressions often involve a mixture of object attributes, spatial relations, and interactions with surrounding entities, which are difficult to capture through a single global similarity score. While CLIP-based similarity provides a strong semantic prior, it is often insufficient to resolve ambiguities in scenes with multiple competing regions. Moreover, object-centric evidence and spatial cues are typically distributed across different regions and scales, requiring explicit mechanisms to expose and integrate such information. Without structured guidance from language parsing and region-level spatial signals, proposal selection becomes highly sensitive to local visual similarity and prone to distraction.

To address the above challenges, we revisit proposal selection in RIS as a robustness-aware contextual search problem. Instead of selecting a mask through isolated single-step ranking, we evaluate whether a candidate remains reliable when visually similar proposals are introduced as distractor contexts. Based on this formulation, we propose RefChess, a training-free framework that searches over sampled distractor subsets and uses Monte-Carlo Tree Search as a practical budgeted mechanism for exploring this combinatorial context space. By integrating language structure, vision–language similarity, object-centric evidence, and spatial guidance into a unified stability-aware reward, RefChess favors proposals that maintain consistent semantic alignment under contextual perturbations, enabling more robust and interpretable referring image segmentation.

Extensive experimental analysis further demonstrates that explicitly modeling contextual interactions and integrating structured semantic cues lead to more stable and reliable proposal selection in zero-shot RIS. The contributions of this paper can be summarized as follows:

- We formulate proposal selection in zero-shot RIS as robustness-aware contextual search over distractor subsets, moving beyond independent proposal ranking.
- We propose RefChess, a training-free framework that instantiates this contextual search with Monte-Carlo Tree Search as a budgeted mechanism for exploring combinatorial distractor contexts.
- We design a stability-aware reward that integrates language decomposition, vision–language similarity, object-centric cues, and spatial guidance, enabling reliable and interpretable proposal selection without task-specific training.

## 2 Related Work

### 2.1 Referring Image Segmentation

Referring image segmentation is a vision–language grounding task that aims to segment a target region in an image according to a natural language expression (Hu et al., 2016).

Unlike object detection or semantic segmentation, RIS requires fine-grained alignment between visual regions and linguistic descriptions, often involving object attributes, spatial relations, and interactions with surrounding entities. Many works on referring image segmentation (Huang & Satoh, 2023; Xu et al., 2023; Chng et al., 2024; Li et al., 2024b;a; Liu et al., 2024; Nag et al., 2024; Shah et al., 2024; Shang et al., 2024; Huang et al., 2025) mainly adopt a fully supervised setting, where models are trained with pixel-level masks paired with referring expressions. Although effective, such approaches rely on large-scale dense annotations, which are expensive to collect and limit scalability and generalization (Wang et al., 2024a;b). To reduce annotation cost, weakly supervised and semi-supervised methods (Lee et al., 2023b; Liu et al., 2023; Dai & Yang, 2024; Yu et al., 2024; Liu et al., 2025) have been explored using cheaper forms of supervision such as bounding boxes or coarse annotations. However, these methods still require task-specific training and provide limited guidance for precise region localization, leaving generalization to complex scenes and diverse expressions largely unresolved (Ni et al., 2024; Liu & Li, 2025).

### 2.2 General-Purpose Models for Image Segmentation

Recent progress in image segmentation has been driven by general-purpose foundation models (Bommasani et al., 2022). The SAM supports prompt-driven, class-agnostic mask generation and provides high-quality instance-level proposals with strong generalization ability, making it widely adopted as a generic proposal generator in downstream segmentation and grounding tasks (Cheng et al., 2023; Zou et al., 2023; Chen et al., 2024; Gundavarapu et al., 2024). Although SAM 3 (Carion et al., 2025) supports text prompts for concept segmentation, it relies on extensive phrase–mask supervision and is mainly geared toward short noun phrases rather than reasoning-heavy referring expressions. Therefore, to keep our setting training-free and reproducible, we use SAM as a class-agnostic proposal generator and focus on robust, language-aware selection among visually similar candidates (Ma et al., 2024; Zhang et al., 2025).

Vision–language models such as CLIP learn aligned representations of images and text, enabling open-vocabulary semantic matching for training-free scoring or ranking of candidate regions. Despite their effectiveness across many vision–language tasks (Yu et al., 2023; Li et al., 2024c), CLIP-based similarity is typically computed independently for each region and remains insensitive to contextual interactions or relational reasoning (Lewis et al., 2024; Patel et al., 2024). Overall, while SAM and CLIP form a strong foundation for zero-shot segmentation, their limitations motivate more structured and context-aware proposal selection strategies.

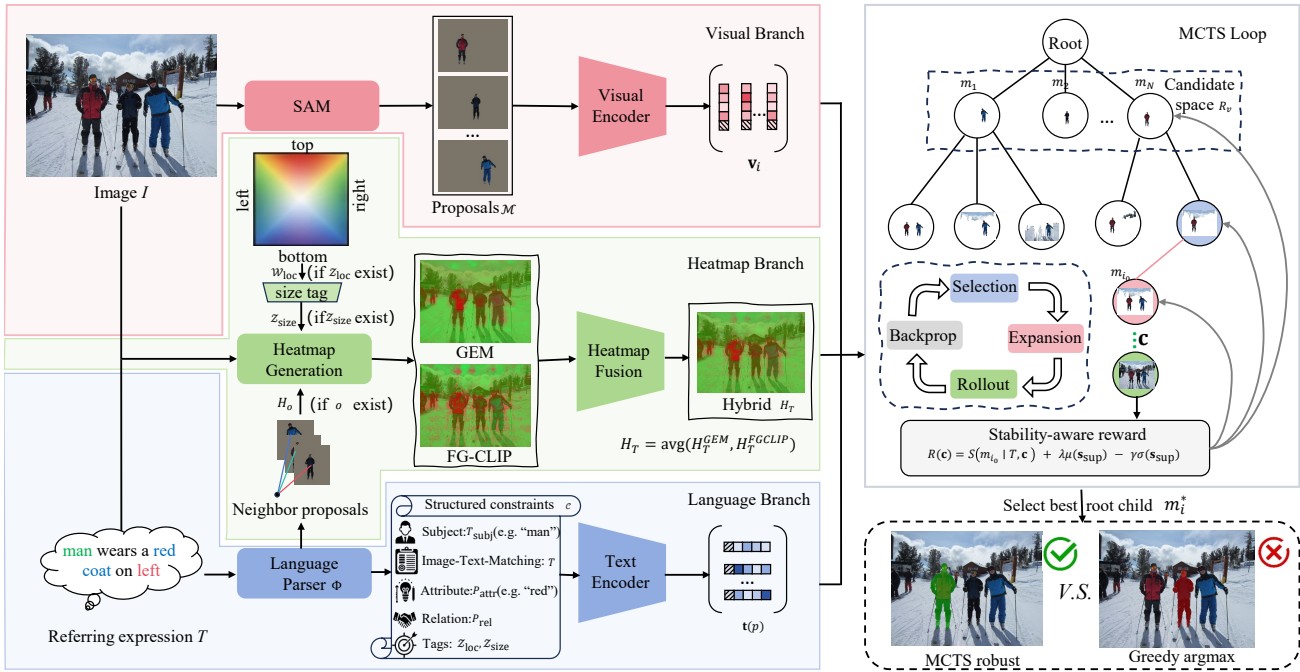

*Figure 2.* Overview of RefChess. We generate SAM mask proposals and score them with CLIP-based visual/language cues and hybrid heatmaps (GEM + FG-CLIP). An MCTS selector then chooses the most stable proposal under context perturbations, improving robustness over greedy top-1 selection.

## 2.3  Zero-shot Referring Image Segmentation

Zero-shot referring image segmentation addresses the scalability limits of supervised approaches by leveraging pretrained foundation models. A dominant formulation treats zero-shot RIS as a proposal ranking problem, where candidate regions are aligned with referring expressions using vision–language similarity. Methods such as Global-Local (Yu et al., 2023) and CaR (Sun et al., 2024) follow this paradigm, focusing on global–local feature aggregation or iterative refinement to improve alignment quality.

With the introduction of the SAM, recent work shifts attention toward enhancing region–text alignment given high-quality, class-agnostic proposals. Along this line, HybridGL (Liu & Li, 2025) improves mask representations through hybrid global–local features and spatial guidance, while TAS (Suo et al., 2023) and PseudoRIS (Yu et al., 2024) incorporate additional semantic context from captioning models. Despite these advances, most zero-shot RIS methods continue to evaluate candidate regions in isolation, implicitly assuming that the correct proposal can be identified independently.

## 3  Method

### 3.1  Overview

Given an input image $I \in \mathbb{R}^{H \times W \times 3}$ and a referring expression $T$, referring image segmentation aims to predict a binary mask $m_* \in \{0, 1\}^{H \times W}$. As illustrated in

Fig. 2, RefChess first uses a pretrained Segment Anything Model (SAM) to generate class-agnostic mask proposals $\mathcal{M} = \{m_1, \ldots, m_N\}$. Each proposal is encoded by frozen CLIP visual encoders, while $T$ is encoded by the CLIP text encoder. We then parse $T$ into structured constraints $\mathcal{C}$ (subject/attribute/relation prompts with optional proximity and location/size cues) and compute a unified proposal score $S(m_i \mid T, \mathcal{C})$ by combining CLIP-based region–text compatibility with lightweight heatmap priors.

Instead of selecting $\arg\max_i S(m_i \mid T, \mathcal{C})$, RefChess applies Monte Carlo Tree Search (MCTS) to model random perturbation contexts. MCTS simulates short chains where the first proposal is the primary candidate and the rest act as distractors, and evaluates them with a stability-aware reward. The final mask is chosen as the root child with the highest empirical mean value, yielding a robustness-aware selection for zero-shot referring image segmentation.

### 3.2  Proposal Generation and Representation

In this section, we generate object-level mask proposals with SAM and encode each proposal into a CLIP visual feature for subsequent language-driven scoring.

We use SAM to generate mask proposals $\mathcal{M} = \{m_i\}_{i=1}^{N}$ from the input image $I$, where each $m_i$ denotes a candidate object region. For CLIP encoding, we retain the masked foreground and fill the background with the CLIP pixel mean (Radford et al., 2021). The processed proposals $m_i$ are passed through the CLIP visual encoder to extract their

feature vectors. For each processed proposal, the feature vector $\mathbf{v}_i \in \mathbb{R}^d$ is obtained:

$$\mathbf{v}_i = \text{CLIP}_{\text{visual}}\,(m_i), \qquad (1)$$

These feature vectors $\mathbf{v}_i$ capture the visual semantics of each proposal, allowing us to align them with the referring expression in the next step.

### 3.3 Language-Guided Proposal Scoring

In this section, we propose an interpretable, training-free *constraint-aware* scoring function that converts the referring expression into structured constraints and unifies CLIP region–text similarity with minimal spatial priors, enabling more reliable grounding among competing SAM proposals.

#### 3.3.1 TEXT DECOMPOSITION AND PROMPT SETS

To integrate free-form referring expressions into our constraint-aware scoring, we pass the expression $T$ through a lightweight parser $\Phi(\cdot)$ to obtain explicit, factorized linguistic constraints $\mathcal{C} = \Phi(T)$, $\mathcal{C}$ including: a subject head noun $T_{\text{subj}}$, an attribute prompt set $\mathcal{P}_{\text{attr}}$, a relation prompt set $\mathcal{P}_{\text{rel}}$, an optional proximity object noun $o$, coarse location tags $\mathcal{Z}_{\text{loc}} \subseteq \{left,right,top,bottom,center\}$, and an optional size preference $\mathcal{Z}_{\text{size}} \in \{big, small\}$.

For example, given "*the man wearing a blue coat on the left*", $\Phi$ yields $T_{\text{subj}} = man$, relation phrases such as "*man wear coat*", and $\mathcal{Z}_{\text{loc}} = \{left\}$. If the expression additionally contains "*next to a car*", then $o = car$ is extracted as a proximity prior.

#### 3.3.2 CLIP SEMANTIC COMPATIBILITY

For each proposal $m_i$, we extract a CLIP embedding $\mathbf{v}_i$ and encode any prompt $p$ into $\mathbf{t}(p)$. The basic region–text compatibility is computed by CLIP-scaled cosine similarity:

$$s(m_i \mid p) = \exp(\tau)\,\frac{\mathbf{v}_i^\top \mathbf{t}(p)}{\|\mathbf{v}_i\|_2\,\|\mathbf{t}(p)\|_2}, \qquad (2)$$

where $\exp(\tau)$ is the CLIP logit scale. We use $s(m_i \mid T_{\text{subj}})$ as the subject term; we average $s(m_i \mid p)$ over $p \in \mathcal{P}_{\text{attr}}$ and $p \in \mathcal{P}_{\text{rel}}$ to obtain attribute and relation terms. In addition, we also maintain a global matching term $s(m_i \mid T)$ (ITM-style) to stabilize cases where local prompts are incomplete.

#### 3.3.3 LOCATION AND SIZE AWARE HEATMAP EVIDENCE

Following prior text-conditioned localization signals (Liu & Li, 2025), we obtain a text-conditioned heatmap $H_T$ from the full expression $T$ using GEM/FG-CLIP (Walid Bousselham, 2024; Xie et al., 2025). We then incorporate explicit location tags $\mathcal{Z}_{\text{loc}}$ by multiplying $H_T$ with a deterministic weight map $\mathcal{W}_{\text{loc}}$ (a 1→0 or 0→1 ramp along the corresponding axis; center uses a separable triangular peak), and define the proposal evidence as:

$$E_{\text{hot}}(m_i) = \frac{1}{|m_i|} \sum_{(x,y)\in m_i} \big(H_T(x,y) \cdot \mathcal{W}_{\text{loc}}(x,y)\big). \quad (3)$$

For size tags $\mathcal{Z}_{\text{size}}$, we apply an area-based reweighting to $E_{\text{hot}}$, but only for proposals whose heatmap response is above the median, which prevents size cues from amplifying irrelevant regions. Importantly, in our framework $E_{\text{hot}}$ is not used as a standalone locator; it is a lightweight, interpretable reward term that complements the language constraints and stabilizes proposal selection under visually similar distractors.

#### 3.3.4 PROXIMITY OBJECT HEATMAP PRIOR WITH NEIGHBORHOOD PROPAGATION

When a proximity object noun $o$ is present, we build a dense text-conditioned heatmap $H_o$ from the original image paired with $o$. We convert it into a proposal-level prior by averaging heatmap values inside each mask, and then apply a local max-propagation over spatial neighbors:

$$
\begin{aligned}
\tilde{E}_{\text{obj}}(m_i) &= \frac{1}{|m_i|} \sum_{(x,y)\in m_i} H_o(x,y), \\
E_{\text{obj}}(m_i) &= \max_{m_j \in \mathcal{N}(m_i)\cup\{m_i\}} \tilde{E}_{\text{obj}}(m_j),
\end{aligned}
\qquad (4)
$$

where $\mathcal{N}(m_i)$ denotes *spatially adjacent proposals*, defined as proposals whose normalized bounding-box centers (*neighbor factor*) lie within a fixed Euclidean radius.

Finally, we design a unified, constraint-aware proposal score that aggregates heterogeneous evidence (language factors, region–text compatibility, object-centric cues, and spatial heatmap evidence) into a single scalar:

$$
\begin{aligned}
S(m_i \mid T, \mathcal{C}) = {} & \lambda_{\text{subj}}\, s(m_i \mid T_{\text{subj}}) + \lambda_{\text{itm}}\, s(m_i \mid T) \\
& + \lambda_{\text{attr}} \operatorname*{avg}_{p \in \mathcal{P}_{\text{attr}}} s(m_i \mid p) + \lambda_{\text{obj}}\, E_{\text{obj}}(m_i) \\
& + \lambda_{\text{rel}} \operatorname*{avg}_{p \in \mathcal{P}_{\text{rel}}} s(m_i \mid p) + \lambda_{\text{hot}}\, E_{\text{hot}}(m_i).
\end{aligned}
$$
$$(5)$$

Where $\lambda_{\mathcal{K}}$, $\mathcal{K} \in \{\text{subj}, \text{attr}, \text{rel}, \text{itm}, \text{obj}, \text{hot}\}$ are fixed weights, and $S(\cdot)$ serves as our *scoring primitive* for the subsequent stability-aware selection stage.

### 3.4 Contextual Search for Robust Proposal Selection

The previous section yields a score $S(m_i \mid T, \mathcal{C})$ for each mask; we denote $S_i \triangleq S(m_i \mid T, \mathcal{C})$. Rather than selecting the final mask by the isolated greedy choice $\arg\max_i S_i$, we evaluate each candidate under sampled distractor contexts. Exhaustively enumerating all distractor subsets is combinatorial, so we use MCTS (Wiechowski et al., 2022) as a budgeted search mechanism to allocate simulations toward informative contextual perturbations and identify the proposal with the most stable contextual compatibility.

Each simulation follows the standard MCTS loop: **Selection** traverses the search tree via an upper-confidence rule, **Ex-**

**pansion** adds a new child by appending an untried proposal index, **Simulation/Rollout** samples additional distractors up to depth $D$ and evaluates the resulting context by $R(\pi)$, and **Backpropagation** propagates the reward to update node statistics for subsequent searches.

### 3.4.1 SEARCH FORMULATION OVER DISTRACTOR CONTEXT

We represent each search state using lightweight index bookkeeping that matches the implementation. Specifically, the search incrementally builds a sampled context $\mathbf{c} = (i_0, i_1, \ldots, i_{L-1})$ of distinct proposal indices, where all indices are sampled *without replacement* ($i_\ell \neq i_{\ell'}$ for $\ell \neq \ell'$). Here, the first element $i_0$ is the *primary* proposal, i.e., the candidate mask to be evaluated, while the remaining elements $\{i_1, \ldots, i_{L-1}\}$ serve as *distractor* proposals. These distractors do *not* change the returned mask; instead, they define sampled contextual perturbations used by the rollout reward to measure whether the primary proposal remains competitive when other plausible regions are considered. Importantly, the sequence $\mathbf{c}$ is used as a search and bookkeeping structure rather than as a temporal decision trajectory.

A node $v$ corresponds to a partial sampled context, denoted by $\mathbf{c}_v = (i_0, \ldots, i_{d-1})$, where $d$ is the current depth. The set of indices that have not been used so far is tracked as the remaining candidate set $\mathcal{R}_v = \{1, \ldots, N\} \setminus \{i_0, \ldots, i_{d-1}\}$. Expanding a node amounts to choosing one unused proposal index $a \in \mathcal{R}_v$ and appending it to the current context. We denote this append operation by $\oplus$, i.e., the successor node $v'$ has $\mathbf{c}_{v'} = \mathbf{c}_v \oplus a$, and its remaining set shrinks accordingly as $\mathcal{R}_{v'} = \mathcal{R}_v \setminus \{a\}$.

Each node maintains the visit count $N(v)$ and the accumulated value $W(v)$ collected from rollouts. Their ratio $Q(v) = \frac{W(v)}{N(v)+\epsilon}$ is the empirical mean value used by the upper confidence bound (UCB) criterion during selection, which balances exploitation (high $Q$) and exploration (low $N$).

### 3.4.2 SELECTION WITH UCB AND EXPANSION

During **Selection**, we traverse the tree using an UCB rule that trades off exploiting high-value children and exploring under-visited ones, thereby allocating simulations adaptively to ambiguous proposals. For node $v$ and child $u \in \text{Child}(v)$:

$$u^\star = \arg\max_{u \in \text{Child}(v)} \left[ \frac{W(u)}{N(u)+\epsilon} + c_{\text{puct}} \sqrt{\frac{\ln(N(v)+1)}{N(u)+\epsilon}} \right]. \tag{6}$$

where $c_{\text{puct}}$ controls the exploration–exploitation trade-off (larger values encourage exploring less-visited children), and $\epsilon > 0$ prevents division by zero and improves numerical

---

**Algorithm 1** MCTS-based Contextual Search for Stability-Aware Proposal Selection

---

**Input:** proposals $\mathcal{M} = \{m_i\}_{i=1}^N$, scores $S_i = S(m_i \mid T, \mathcal{C})$, simulations $M$, rollout depth $D$, exploration $c_{\text{puct}}$, coefficients $(\lambda, \gamma)$.
**Output:** $i^\star$ (return mask $m_{i^\star}$).
Initialize root $v_0$: $\mathbf{c}_{v_0} = \emptyset$, $\mathcal{R}_{v_0} = \{1, \ldots, N\}$, $N(v_0) = W(v_0) = 0$.
**for** $k = 1$ **to** $M$ **do**
  $v \leftarrow v_0$; path $\mathcal{P} \leftarrow [v_0]$. {Selection}
  **while** $v$ has children **and** $v$ is fully expanded **do**
    Select $u \in \text{Child}(v)$ using Eq. (6).
    $v \leftarrow u$; append $v$ to $\mathcal{P}$.
  **end while**{Expansion}
  **if** $\exists$ untried $a \in \mathcal{R}_v$ **then**
    Create $v'$ by $\mathbf{c}_{v'} = \mathbf{c}_v \oplus a$, $\mathcal{R}_{v'} = \mathcal{R}_v \setminus \{a\}$.
    $v \leftarrow v'$; append $v$ to $\mathcal{P}$.
  **end if**{Simulation / Rollout}
  $\mathbf{c} \leftarrow \mathbf{c}_v$; $\mathcal{R} \leftarrow \mathcal{R}_v$.
  **while** $|\mathbf{c}| < D$ **and** $\mathcal{R} \neq \emptyset$ **do**
    Sample $b \sim \text{Unif}(\mathcal{R})$; $\mathbf{c} \leftarrow \mathbf{c} \oplus b$; $\mathcal{R} \leftarrow \mathcal{R} \setminus \{b\}$.
  **end while**{Reward}
  Compute $R(\mathbf{c})$ using Eq. (7). {Backpropagation}
  **for** each node $x$ in $\mathcal{P}$ **do**
    Update $(N(x), W(x))$ using Eq. (8).
  **end for**
**end for**
Return $i^\star$ using Eq. (8).

---

stability when $N(u)$ is small.

If the reached node is not fully expanded, we perform **Expansion** by selecting an untried action $a \in \mathcal{R}_v$ and creating a new child node. A rollout then samples additional indices uniformly without replacement until a depth limit $D$ is reached.

### 3.4.3 STABILITY-AWARE REWARD

During **Simulation**, the rollout approximates the expectation over perturbations, so a proposal is preferred only if it remains consistently favorable under diverse distractor contexts. Given a completed chain $\mathbf{c}$, we evaluate the primary index $i_0$ using both its own score and the stability of perturbation scores. Let $\mathbf{s}_{\text{sup}} = (S_{i_1}, \ldots, S_{i_{L-1}})$. We define the mean, standard deviation, and the final reward jointly:

$$\mu(\mathbf{s}_{\text{sup}}) = \frac{1}{L-1} \sum_{\ell=1}^{L-1} S_{i_\ell},$$

$$\sigma(\mathbf{s}_{\text{sup}}) = \sqrt{\frac{1}{L-1} \sum_{\ell=1}^{L-1} \left( S_{i_\ell} - \mu(\mathbf{s}_{\text{sup}}) \right)^2}, \tag{7}$$

$$R(\mathbf{c}) = S_{i_0} + \lambda \mu(\mathbf{s}_{\text{sup}}) - \gamma \sigma(\mathbf{s}_{\text{sup}}),$$

*Table 1.* Comparisons with SOTA referring image segmentation methods on standard benchmarks. The best and second best results are highlighted with red shading and blue shading, respectively. * indicates the extra dataset used to train the model.

| Metric | Method | Vision Backbone | Pre-trained Model | RefCOCO | | | RefCOCO+ | | | RefCOCOg | |
| --- | --- | --- | --- | --- | --- | --- | --- | --- | --- | --- | --- |
| | | | | val | testA | testB | val | testA | testB | val | test |
| oIoU | *zero-shot methods w/ additional training* | | | | | | | | | | |
| | Pseudo-RIS (Yu et al., 2024) | ViT-B | SAM, CoCa, CLIP | 37.33 | 43.43 | 31.90 | 40.19 | 46.43 | 33.63 | 41.63 | 43.52 |
| | VLM-VG (Wang et al., 2024a) | R101 | COCO*, VLM-VG* | 45.40 | 48.00 | 41.40 | 37.00 | 40.70 | 30.50 | 42.80 | 44.10 |
| | *zero-shot methods w/o additional training* | | | | | | | | | | |
| | Grad-CAM (Selvaraju et al., 2017) | R50 | SAM, CLIP | 23.44 | 23.91 | 21.60 | 26.67 | 27.20 | 24.84 | 23.00 | 23.91 |
| | MaskCLIP (Zhou et al., 2022) | R50 | SAM, CLIP | 20.18 | 20.52 | 21.30 | 22.06 | 22.43 | 24.61 | 23.05 | 23.41 |
| | Global-Local (Yu et al., 2023) | R50 | FreeSOLO, CLIP | 24.58 | 23.38 | 24.35 | 25.87 | 24.61 | 25.61 | 30.07 | 29.83 |
| | Global-Local (Yu et al., 2023) | R50 | SAM, CLIP | 24.55 | 26.00 | 21.03 | 26.62 | 29.99 | 22.23 | 28.92 | 30.48 |
| | Global-Local (Yu et al., 2023) | ViT-B | SAM, CLIP | 21.71 | 24.48 | 20.51 | 23.70 | 28.12 | 21.86 | 26.57 | 28.21 |
| | Ref-Diff (Minheng Ni, 2023) | ViT-B | SAM, SD, CLIP | 35.16 | 37.44 | 34.50 | 35.56 | 38.66 | 31.40 | 38.62 | 37.50 |
| | TAS (Suo et al., 2023) | ViT-B | SAM, BLIP2, CLIP | 29.53 | 30.26 | 28.24 | 33.21 | 38.77 | 28.01 | 35.84 | 36.16 |
| | HybridGL (Liu & Li, 2025) | ViT-B | SAM, CLIP | 41.81 | 44.52 | 38.50 | 35.74 | 41.43 | 30.90 | 42.47 | 42.97 |
| | RefChess (**Ours**) | ViT-B | SAM, FGCLIP | 48.47 | 52.50 | 43.58 | 41.20 | 48.19 | 32.73 | 42.85 | 44.21 |
| mIoU | *weakly-supervised methods* | | | | | | | | | | |
| | CLRL (Lee et al., 2023a) | ViT-B | – | 31.06 | 32.30 | 30.11 | 31.28 | 32.11 | 30.13 | 32.88 | – |
| | PPT (Dai & Yang, 2024) | ViT-B | SAM | 46.76 | 45.33 | 46.28 | 45.34 | 45.84 | 44.77 | 42.97 | – |
| | *zero-shot methods w/ additional training* | | | | | | | | | | |
| | Pseudo-RIS (Yu et al., 2024) | ViT-B | SAM, CoCa, CLIP | 41.05 | 48.19 | 33.48 | 44.33 | 51.42 | 35.08 | 45.99 | 46.67 |
| | VLM-VG (Wang et al., 2024a) | R101 | COCO*, VLM-VG* | 49.90 | 53.10 | 46.70 | 42.70 | 47.30 | 36.20 | 48.00 | 48.50 |
| | *zero-shot methods w/o additional training* | | | | | | | | | | |
| | Grad-CAM (Selvaraju et al., 2017) | R50 | SAM, CLIP | 30.22 | 31.90 | 27.17 | 33.96 | 25.66 | 32.29 | 33.05 | 32.50 |
| | MaskCLIP (Zhou et al., 2022) | R50 | SAM, CLIP | 25.62 | 26.66 | 25.17 | 27.49 | 28.49 | 30.47 | 30.13 | 30.15 |
| | Global-Local (Yu et al., 2023) | R50 | FreeSOLO, CLIP | 26.70 | 24.99 | 26.48 | 28.22 | 26.54 | 27.86 | 33.02 | 33.12 |
| | Global-Local (Yu et al., 2023) | R50 | SAM, CLIP | 31.83 | 34.93 | 28.64 | 34.97 | 37.11 | 30.61 | 40.66 | 40.94 |
| | Global-Local (Yu et al., 2023) | ViT-B | SAM, CLIP | 33.12 | 36.52 | 29.58 | 35.29 | 39.58 | 31.89 | 40.08 | 40.74 |
| | CaR (Sun et al., 2024) | ViT-B/ViT-L | CLIP | 33.57 | 35.36 | 30.51 | 34.22 | 36.03 | 31.02 | 36.67 | 36.57 |
| | Ref-Diff (Minheng Ni, 2023) | ViT-B | SAM, SD, CLIP | 37.21 | 38.40 | 37.19 | 37.29 | 40.51 | 33.01 | 44.02 | 44.51 |
| | TAS (Suo et al., 2023) | ViT-B | SAM, BLIP2, CLIP | 39.84 | 41.08 | 36.24 | 43.63 | 49.13 | 36.54 | 46.62 | 46.80 |
| | HybridGL (Liu & Li, 2025) | ViT-B | SAM, CLIP | 49.48 | 53.37 | 45.19 | 43.40 | 49.13 | 37.17 | 51.25 | 51.59 |
| | RefChess (**Ours**) | ViT-B | SAM, FGCLIP | 55.19 | 57.94 | 50.04 | 48.63 | 54.64 | 39.63 | 50.63 | 51.32 |

if $L = 1$ (no perturbation), we simply set $R(\mathbf{c}) = S_{i_0}$. The first term measures the standalone compatibility of the primary proposal, the mean term reflects its contextual competitiveness with respect to sampled distractors, and the standard-deviation penalty discourages proposals whose relative advantage is unstable across different contexts.

### 3.4.4 BACKPROPAGATION

During **Backpropagation**, the rollout outcomes are accumulated to refine empirical values of visited nodes. After obtaining reward $R(\mathbf{c})$, we update all nodes on the visited path $\mathcal{P}$ and finally output the root child with the maximal empirical mean value:

$$N(v) \leftarrow N(v) + 1, \quad W(v) \leftarrow W(v) + R(\mathbf{c}),$$

$$i^{\star} = \arg\max_{u \in \text{Child}(v_0)} \frac{W(u)}{N(u) + \epsilon}, \quad \text{and return } m_{i^{\star}}. \tag{8}$$

Algorithm 1 summarizes our MCTS-based contextual search procedure for selecting proposals that remain stable under sampled distractor contexts.

## 4 Experiments

### 4.1 Datasets and Metrics

**Datasets.** We evaluate RefChess on three standard benchmarks for referring image segmentation, namely RefCOCO (Nagaraja et al., 2016), RefCOCO+ (Nagaraja et al., 2016), and RefCOCOg (Mao et al., 2016). These datasets are built upon MSCOCO (Lin et al., 2014) images and provide each target instance with pixel-level masks paired with referring expressions. RefCOCO typically contains short expressions and often includes explicit spatial words. RefCOCO+ removes such location terms, which reduces the effectiveness of simple positional heuristics. RefCOCOg features longer and more descriptive sentences, placing more emphasis on relational language understanding.

**Metrics.** For evaluation, we adopt two widely used metrics, overall IoU (oIoU) and mean IoU (mIoU). By aggregating intersections and unions over the entire evaluation set before computing the overlap score, oIoU is more sensitive to large error regions and boundary leakage. In contrast, mIoU computes IoU for each instance and then averages

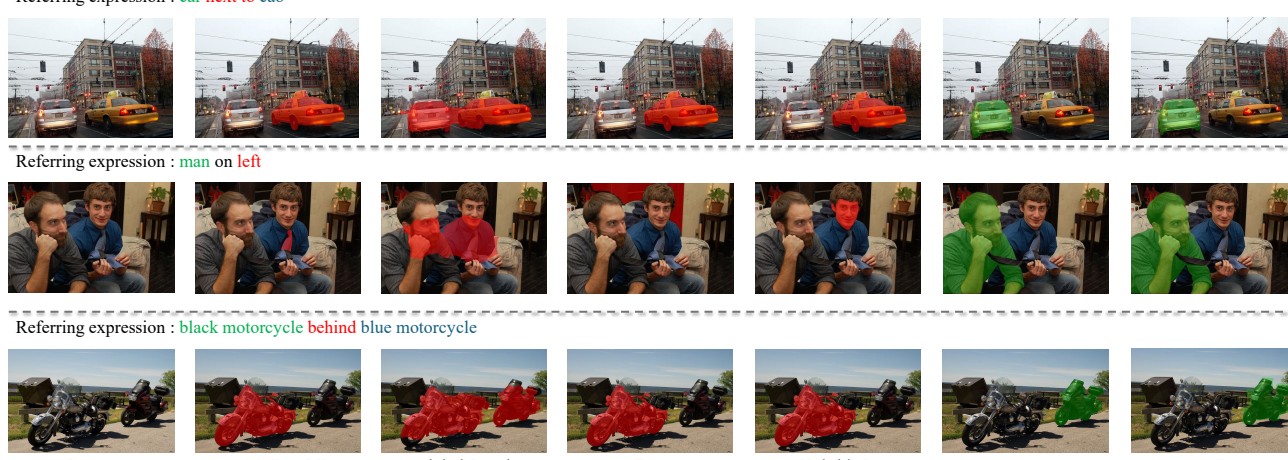

*Figure 3.* Qualitative comparison with representative baselines on challenging referring expressions. Our method more reliably pinpoints the intended instance and yields cleaner, more complete segmentation masks that better match the target object.

across expressions, which prevents large objects from dominating the final score and yields a more balanced reflection of performance across object scales. We report both metrics in all quantitative comparisons.

### 4.2 Implementation Details

All experiments were conducted on one NVIDIA RTX 4090. Following previous works (Minheng Ni, 2023; Suo et al., 2023; Liu & Li, 2025), we generate proposals with SAM using a $8 \times 8$ point grid. The "predicted IoU threshold" and "stability score threshold" are both set to 0.7. Each proposal keeps its foreground and replaces the background with the CLIP pixel mean, then we extract frozen CLIP visual features at $224 \times 224$ proposals. For the heatmap evidence term, we adopt the hybrid backend by default, which averages GEM heatmaps and dense maps produced by a FG-CLIP-base model.

In the unified proposal scoring, we set the four language-alignment weights (subj/attr/rel/itm) to 0.15 each, and assign 10.0 to both the object-prior and heatmap terms to emphasize localization. We define *neighbor factor* (*NF*) with a normalized center-distance radius of 0.3. MCTS uses 2048 simulations with rollout depth 10 and $c_{\text{puct}} = 15$, while the stability-aware reward fixes $\lambda = 0.1$ (support bonus) and $\gamma = 0.05$ (variance penalty) across all datasets.

### 4.3 Results

We compare RefChess with state-of-the-art (SOTA) zero-shot methods on the RefCOCO, RefCOCO+, and Ref-COCOg datasets. As shown in Table 1, RefChess outperforms existing methods across all three benchmarks. Specifically, on RefCOCO and RefCOCO+, RefChess exceeds the SOTA methods by approximately **+2%–8%** in both oIoU

and mIoU across the standard splits. On RefCOCOg, Re-fChess achieves **50.63%** mIoU, falling just **0.62%** short of the best val score, while its oIoU is also among the top results. Compared with weakly-supervised methods (e.g., CLRL and PPT), RefChess is still strong despite being zero-shot: it improves mIoU by **+3.8%–12.6%** on RefCOCO and by **+7.7%** on RefCOCOg (val). On RefCOCO+, it also surpasses PPT by **+3.3%** (val) and **+8.8%** (testA) in mIoU, while being lower on the hardest split (testB). In particular, these improvements are achieved without the need for additional training or external datasets. In contrast, methods like TAS and Ref-Diff rely on larger auxiliary models (e.g., BLIP2 or Stable-Diffusion), adding complexity to the model.

Notably, RefChess also maintains competitive inference efficiency (see Appendix A.1 for details). Additionally, Fig. 3 further supports these quantitative results. RefChess consistently generates cleaner masks and is more accurate in identifying the correct object in complex, multi-object scenes. Its performance is most pronounced when spatial relationships are crucial, such as left/right, front/back, and next-to/behind.

### 4.4 Ablation Study

To evaluate each design choice in RefChess, we conduct extensive ablation studies on the *val* splits of RefCOCO, RefCOCO+, and RefCOCOg.

#### 4.4.1 ABLATION STUDY ON CONTEXTUAL SEARCH STRATEGY

This study analyzes whether contextual search provides benefits beyond one-step proposal ranking, and how sensitive the search is to key MCTS hyperparameters. For a fair com-

*Table 2.* Ablation study on proposal selection, comparing Greedy and MCTS with varying simulation budgets (sims) and rollout depth (Depth $D$) on the *val* splits.

| Setting | RefCOCO | | RefCOCO+ | | RefCOCOg | |
|---|---|---|---|---|---|---|
| | mIoU | oIoU | mIoU | oIoU | mIoU | oIoU |
| Greedy | 40.77 | 29.95 | 46.30 | 35.83 | 46.37 | 34.49 |
| sims=256 | 53.84 | 47.02 | 46.30 | 39.05 | 48.65 | 41.00 |
| sims=1024 | 53.85 | 47.06 | 46.28 | 39.00 | 48.65 | 41.02 |
| sims=4096 | 55.18 | 48.44 | 48.63 | **41.21** | **50.64** | 42.85 |
| $D$=5 | 54.68 | 46.84 | 47.37 | 39.26 | 49.78 | 40.87 |
| $D$=15 | 54.70 | 46.85 | 47.38 | 39.28 | 49.81 | 40.96 |
| default | **55.19** | **48.47** | **48.63** | 41.20 | 50.63 | **42.85** |

*Table 3.* Ablation study on heatmap methods, neighbor factor (*NF*), and size bias on the *val* splits.

| Setting | RefCOCO | | RefCOCO+ | | RefCOCOg | |
|---|---|---|---|---|---|---|
| | mIoU | oIoU | mIoU | oIoU | mIoU | oIoU |
| GEM-based | 53.81 | 47.00 | 46.29 | 39.01 | 48.63 | 40.98 |
| FGCLIP-based | 53.47 | 46.62 | 47.07 | 39.93 | 50.64 | 42.89 |
| *NF*=0.1 | 53.65 | 46.84 | 45.95 | 38.75 | 47.99 | 40.33 |
| *NF*=0.5 | 53.87 | 47.07 | 46.30 | 39.03 | 48.82 | 41.15 |
| w/o size bias | 53.81 | 46.75 | 46.17 | 38.56 | 49.10 | 41.58 |
| default | **55.19** | **48.47** | **48.63** | **41.20** | **50.63** | **42.85** |

parison, the Greedy baseline and MCTS-based selection use the same unified proposal score $S_i$ defined in Eq. (5). Greedy directly selects $\arg\max_i S_i$, while MCTS evaluates candidates under sampled distractor contexts before selecting the final proposal. We vary (i) the number of simulations (sims) and (ii) the rollout depth $D$, and report results on the *val* splits of RefCOCO, RefCOCO+, and RefCOCOg using both mIoU and oIoU.

As shown in Table 2, replacing one-step Greedy selection with contextual search yields substantial improvements across the datasets. Since both variants share the same scoring function, these gains isolate the effect of the search strategy rather than changes in the proposal score. Even with a modest simulation budget (sims=256), the method retains clear improvements over Greedy, suggesting that evaluating candidate proposals under distractor contexts is beneficial.

Further increasing the simulation budget or rollout depth brings only moderate additional gains, while increasing computational cost. We therefore use the default configuration in the remaining experiments, which provides a favorable trade-off between accuracy and efficiency.

### 4.4.2 ABLATION STUDY ON HEATMAP SOURCE AND NEIGHBOR FACTOR PARAMETERS

In this ablation study, we examine the impact of different heatmap generation methods and the *neighbor factor* hyperparameter on the performance of RefChess. Specifically,

*Table 4.* Ablation study on reward components, analyzing the impact of removing individual terms on the *val* splits.

| Setting | RefCOCO | | RefCOCO+ | | RefCOCOg | |
|---|---|---|---|---|---|---|
| | mIoU | oIoU | mIoU | oIoU | mIoU | oIoU |
| $\lambda_{\text{hot}} = 0$ | 53.96 | 46.16 | 46.53 | 38.38 | 49.41 | 40.77 |
| $\lambda_{\text{obj}} = 0$ | 55.19 | 48.45 | 48.49 | 41.06 | **50.64** | **42.96** |
| $\lambda_{\text{rel}} = 0$ | 55.15 | 48.40 | 48.45 | 40.94 | 50.44 | 42.89 |
| $\lambda_{\text{attr}} = 0$ | 54.85 | 48.24 | 48.28 | 41.05 | 50.24 | 42.73 |
| $\lambda_{\text{itm}} = 0$ | 51.72 | 45.99 | 45.26 | 38.72 | 46.98 | 39.66 |
| $\lambda_{\text{subj}} = 0$ | 52.76 | 46.76 | 48.11 | **41.30** | 49.80 | 42.36 |
| default | **55.19** | **48.47** | **48.63** | 41.20 | 50.63 | 42.85 |

we compare the use of GEM-based heatmaps and FG-CLIP-based heatmaps, varying the *neighbor factor* between 0.1 and 0.5. We also analyze the effect of disabling the size bias mechanism, which incorporates size priors into the proposal scoring process.

As shown in Table 3, heatmap design affects the final performance, but it is not the sole source of improvement. FG-CLIP and GEM provide complementary localization cues, and the hybrid design achieves the best overall trade-off across the benchmarks. The *neighbor factor* (*NF*) also affects performance, with moderate values providing more stable behavior than overly narrow or overly broad neighborhoods.

Disabling the size bias term leads to performance drops, particularly on RefCOCO+, suggesting that coarse object-size priors provide useful supplementary evidence in multi-object scenes. We emphasize that these spatial and object-centric cues act as stabilizers for proposal selection, while language-based matching remains the primary semantic signal.

### 4.4.3 ABLATION STUDY ON EFFECT OF REWARD COMPONENTS

In this ablation study, we investigate the impact of different components of the unified proposal score used by RefChess. The components correspond to full-expression matching, subject cues, attribute cues, relation cues, object-centric evidence, and spatial heatmap guidance. We ablate each component by setting its corresponding coefficient to zero and evaluate performance on the *val* splits of the three benchmarks.

It is important to note that coefficient magnitudes are mainly used for scale alignment across heterogeneous signals and should not be directly interpreted as semantic importance. The CLIP-based language similarities and the spatial/object-centric scores have different numerical ranges, so the weights normalize their contributions to comparable scales.

As shown in Table 4, language-related components remain

essential. Removing the full-expression or subject-related terms leads to clear performance drops, especially on RefCOCO, indicating that language matching provides the primary discriminative signal for identifying the referred target. Object-centric and heatmap cues also contribute, but they are better understood as supplementary stabilizers that help resolve ambiguous or cluttered scenes. Overall, the default configuration provides the most stable trade-off across datasets.

## 5 Conclusion

This paper proposes RefChess, a training-free contextual search framework for zero-shot referring image segmentation. RefChess addresses the brittleness of independent proposal ranking by evaluating candidate masks under sampled distractor contexts, with MCTS serving as a budgeted search mechanism for exploring the resulting combinatorial space. Combined with language-aware cue decomposition, object-centric evidence, and spatial guidance, the proposed stability-aware selection strategy achieves consistent improvements on RefCOCO, RefCOCO+, and RefCOCOg without task-specific training. Future work will focus on stronger proposal priors, tighter language–mask interaction, and extensions from single-instance referring segmentation to set-valued or semantic queries.

## Impact Statement

RefChess is a training-free framework for zero-shot referring image segmentation that improves robustness under ambiguity by casting proposal selection as a decision-making problem and using MCTS to resist distractors. It can reduce reliance on labeled data and make language-guided segmentation more reliable for human–computer interaction and robotics.

Potential risks include biases inherited from foundation models and misuse for privacy-invasive surveillance; high-stakes deployments should include bias auditing, uncertainty communication, and use restrictions.

## Acknowledgments

This work was supported by National Natural Science Fund of China (Grant number 62573123), Research Fund for Advanced Ocean Institute of Southeast University, Nantong (GP202411), and the Fundamental Research Funds for the Central Universities. We also thank the Big Data Computing Center of Southeast University for providing the facility support on the numerical calculations in this paper.

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

# A Appendix

## A.1 Complexity Analysis

We report the inference time with SOTA zero-shot RIS methods. All timings are measured on a single NVIDIA RTX 4090 with a single $224 \times 224$ input image and RefCOCO (val), and summarized in Table 5 and Table 6. Specifically, Table 5 compares the inference speed with representative baselines, while Table 6 analyzes the runtime impact of different heatmap designs and the MCTS simulation budget.

*Table 5.* Inference speed (ms) and accuracy comparison with representative baselines.

| Method | Global-Local (FreeSOLO) | Global-Local (SAM-R50) | Global-Local (SAM-ViT-B) | TAS | HybridGL | Ours |
|---|---|---|---|---|---|---|
| **Time (ms)** | 284.47 | 541.42 | 612.32 | 2593.87 | 862.43 | 1231.21 |
| **mIoU** | 26.70 | 31.83 | 33.12 | 39.84 | 49.48 | 55.19 |
| **oIoU** | 24.58 | 24.55 | 21.71 | 29.53 | 41.81 | 48.47 |

*Table 6.* Inference speed (ms) and accuracy comparison under different settings.

| Setting | w/o heatmap | GEM heatmap | FGCLIP heatmap | sim=256 | sim=1024 | sim=4096 | $D$=5 | $D$=15 | default |
|---|---|---|---|---|---|---|---|---|---|
| **Time (ms)** | 663.31 | 696.28 | 1059.03 | 842.34 | 1077.21 | 1321.19 | 1145.84 | 1289.33 | 1231.21 |
| **mIoU** | 53.96 | 53.81 | 53.47 | 53.84 | 53.85 | 55.18 | 54.68 | 54.70 | 55.19 |
| **oIoU** | 46.16 | 47.00 | 46.62 | 47.02 | 47.06 | 48.44 | 46.84 | 46.85 | 48.47 |

We compare inference latency and accuracy with representative baselines in Table 5. RefChess is slower than the lightest Global-Local (Yu et al., 2023) variants, but achieves substantially higher segmentation quality, improving mIoU from 33.12 to 55.19 and oIoU from 21.71 to 48.47 over Global-Local (SAM-ViT-B). Compared with stronger zero-shot systems, RefChess delivers the best accuracy among all listed methods while keeping runtime in a practical range: it is markedly faster than TAS (Suo et al., 2023) and achieves higher mIoU/oIoU than HybridGL (Liu & Li, 2025), indicating a favorable accuracy–latency trade-off. To further contextualize our method's validity, Table 6 reports the runtime impact of key design choices. Adding a heatmap incurs moderate overhead: GEM increases latency from 663 to 696 ms, while FGCLIP is noticeably heavier. Although either heatmap alone can be brittle (often highlighting coarse or biased regions), combining GEM and FGCLIP provides complementary localization cues; their agreement is more consistent across distractors, which yields a clear accuracy gain when used as heatmap evidence in our scoring.

For MCTS, latency increases with the simulation budget as expected, and larger budgets achieve the best accuracy. Increasing rollout depth $D$ further increases runtime, but brings negligible accuracy change, suggesting that allocating computation to more simulations is generally more cost-effective than using deeper rollouts under our default setup. Notably, even without heatmap guidance, RefChess runs faster than HybridGL while achieving higher accuracy. This indicates that the main performance gains of RefChess are not solely attributed to the heatmap module, and the proposed robust proposal selection via MCTS provides strong accuracy improvements with practical inference cost.

## A.2 More Ablation Studies

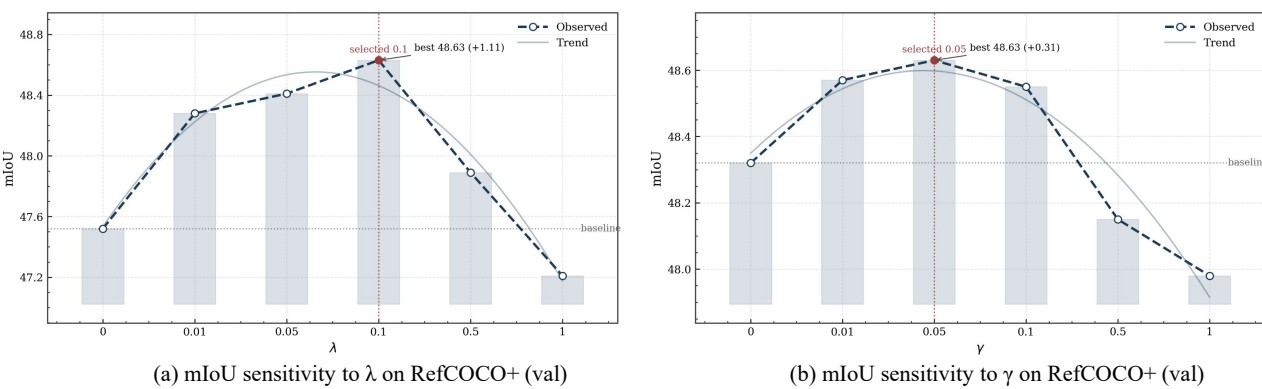

(a) mIoU sensitivity to $\lambda$ on RefCOCO+ (val)      (b) mIoU sensitivity to $\gamma$ on RefCOCO+ (val)

*Figure 4.* Sensitivity of mIoU to hyper-parameters on RefCOCO+ (val); (a) $\lambda$ with the best at $\lambda$=0.1; (b) $\gamma$ with the best at $\gamma$=0.05.

We further study the sensitivity of two scalar hyper-parameters $\lambda$ and $\gamma$ on RefCOCO+ (val). As shown in Fig. 4, varying $\lambda$ exhibits a clear unimodal trend, where performance improves from $\lambda=0$ to a peak at $\lambda=0.1$ (48.63 mIoU) and then degrades when $\lambda$ becomes large. A similar pattern is observed for $\gamma$, with the best performance achieved around $\gamma=0.05$, while overly large values lead to a gradual drop. Overall, the method is stable for small-to-moderate ranges, and we use $\lambda=0.1$ and $\gamma=0.05$ as default settings.

## A.3 Limitations

Despite improved robustness in proposal selection, RefChess still fails in several challenging cases, as shown in Fig. 5. First, like most proposal-based zero-shot RIS methods, its performance is ultimately bounded by the quality and coverage of the proposal set. If the true instance is missing from the candidates, or only appears as a coarse or fragmented mask, no downstream selection strategy can recover an accurate segmentation. Second, expressions requiring fine-grained grounding, such as part-level references, subjective or implicit attributes, or long relational descriptions with multiple context entities, remain challenging. In such cases, multiple visually plausible candidates may partially satisfy the expression, weakening the discriminability of language and spatial reward signals and occasionally leading to incorrect selections, especially in cluttered scenes or for small targets. Third, RefChess is designed for standard single-instance RIS benchmarks, where each expression refers to one target mask. Extending the framework to set-valued or semantic queries, such as segmenting all instances of a category, would require modifying the objective from single-proposal selection to set prediction, which we leave for future work.

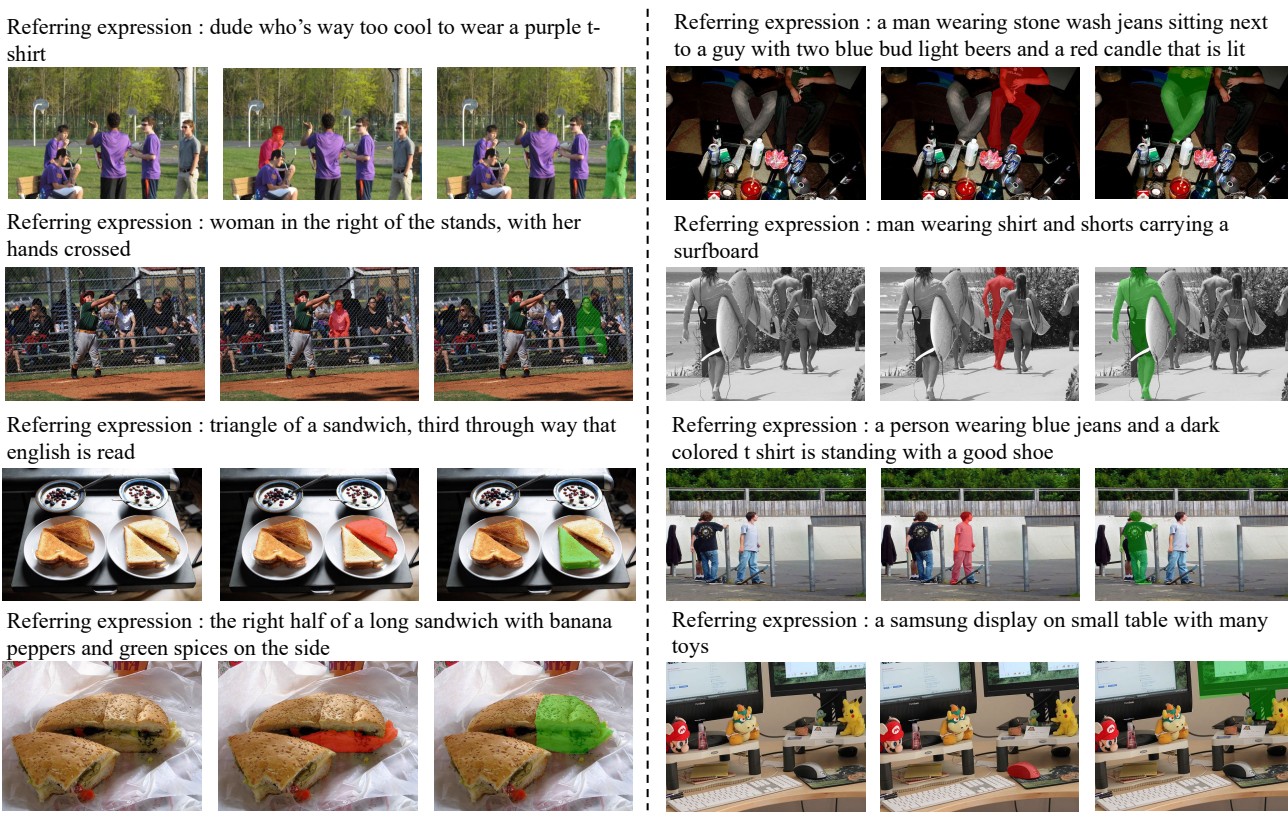

Referring expression : dude who's way too cool to wear a purple t-shirt

Referring expression : a man wearing stone wash jeans sitting next to a guy with two blue bud light beers and a red candle that is lit

Referring expression : woman in the right of the stands, with her hands crossed

Referring expression : man wearing shirt and shorts carrying a surfboard

Referring expression : triangle of a sandwich, third through way that english is read

Referring expression : a person wearing blue jeans and a dark colored t shirt is standing with a good shoe

Referring expression : the right half of a long sandwich with banana peppers and green spices on the side

Referring expression : a samsung display on small table with many toys

| Image | Bad case | GT | Image | Bad case | GT |

*Figure 5.* Representative failure cases of our method. Columns show the input image, our incorrect prediction (*Bad case*), and the ground-truth mask (*GT*). Failures commonly occur when the target is not well covered by proposals, when the expression involves fine-grained parts or implicit attributes, and when complex relations in cluttered scenes yield multiple plausible candidates.

## A.4 More Visual Comparisons

We provide additional qualitative comparisons on RefCOCO, RefCOCO+, and RefCOCOg (val) in Fig. 6–8. Across diverse referring expressions involving attributes, spatial relations, and multi-instance ambiguity, our method more reliably identifies the intended target and produces cleaner, more complete masks than prior zero-shot baselines.

Referring expression : white horse in back left of kid in red

Referring expression : sitting back to us

Referring expression : person on the right

Referring expression : green right

Referring expression : smallest yellow cylinder

Referring expression : small one

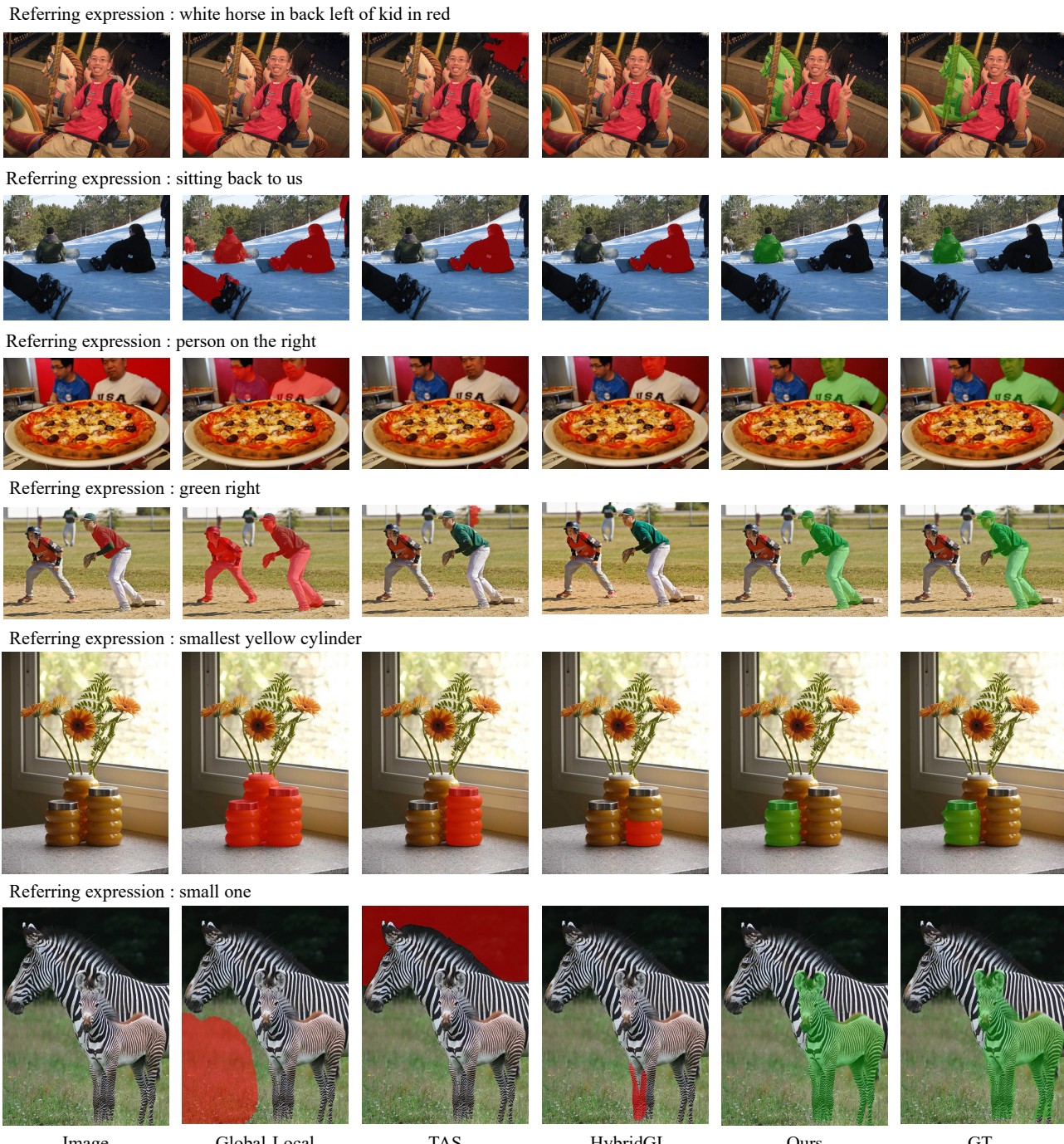

| Image | Global-Local | TAS | HybridGL | Ours | GT |

*Figure 6.* Qualitative comparisons on RefCOCO (val). Our method better localizes the target instance and yields more complete masks under multi-instance ambiguity and spatial descriptions.

Referring expression : black jacket pants by the child

Referring expression : closest girl in blue head turned

Referring expression : piece at 12 o clock

Referring expression : purple sofa next to lamp

Referring expression : elephant behind bush

Referring expression : shorter person

Referring expression : cup neares edge

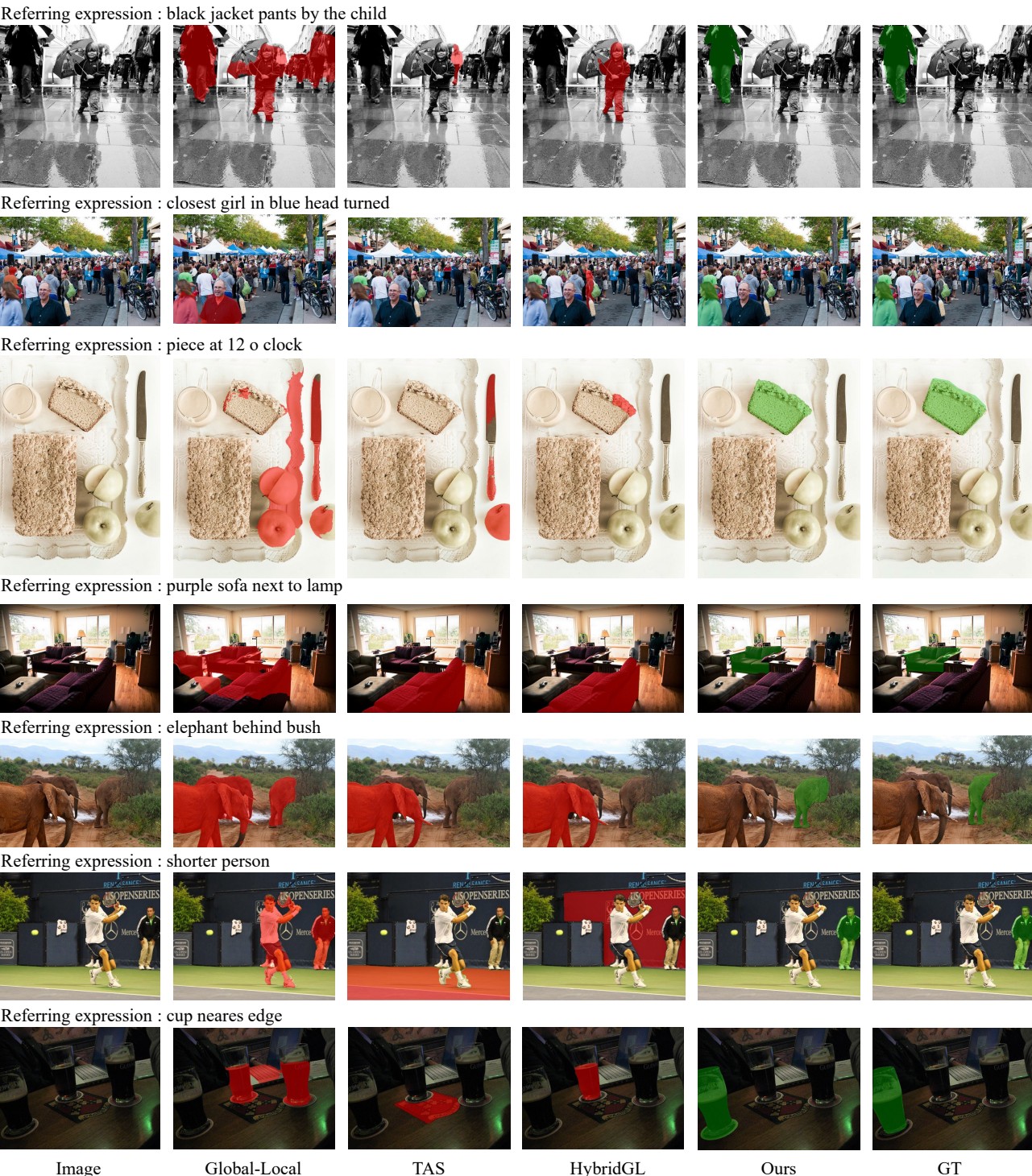

| Image | Global-Local | TAS | HybridGL | Ours | GT |

*Figure 7.* Qualitative comparisons on RefCOCO+ (val). Our method is more robust to attribute-centric expressions and challenging scenes with similar candidates, producing masks that better match the referred object.

Referring expression : cow on the far right who is barely visible

Referring expression : catcher and umpire

Referring expression : the rightmost spoon

Referring expression : cat bed

Referring expression : a light blue backpack on the lap of the passenger next to the one using his cell phone

Referring expression : body of person standing in the corner of the pic

Referring expression : left most white umbrella

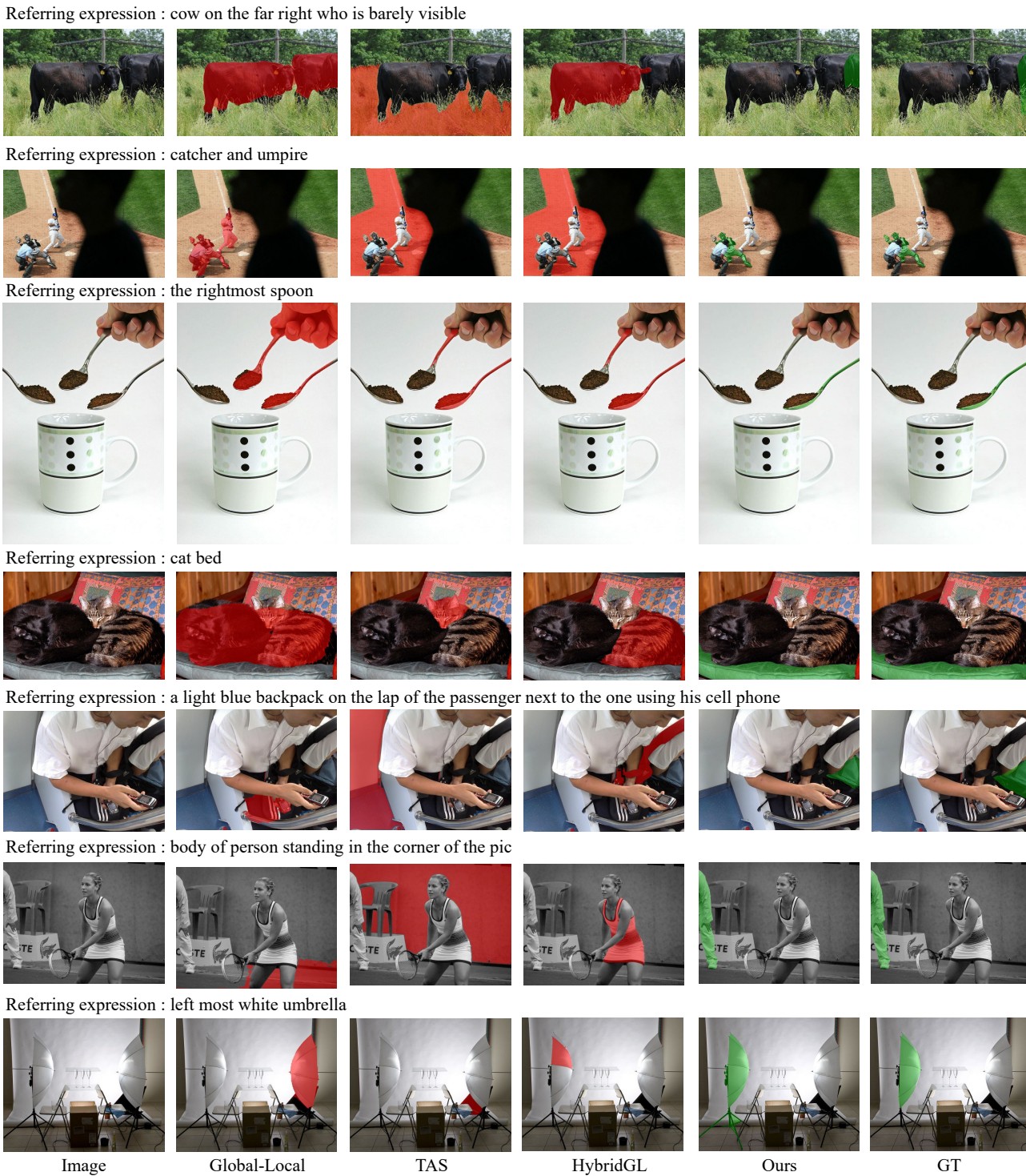

| Image | Global-Local | TAS | HybridGL | Ours | GT |

*Figure 8.* Qualitative comparisons on RefCOCOg (val). Our method improves grounding for longer expressions with richer relations and context, leading to cleaner segmentation and fewer distractor regions.

