# OpenReview forum: "RefChess: Training-Free Contextual Search for Zero-Shot Referring Image Segmentation"
_ICML.cc/2026/Conference — ICML 2026 regular_

### Official Review · Reviewer_ynba · 2026-02-27

**Soundness:** 2
**Presentation:** 1
**Significance:** 2
**Originality:** 3
**Overall Recommendation:** 3
**Confidence:** 4

**Summary:**

This paper introduces RefChess, a training-free framework for zero-shot RIS task. The method reformulates mask selection as a decision-making process using Monte Carlo Tree Search (MCTS). To guide this search process, the model designs a composite score that integrates fine-grained language-aware cue decomposition and multiple visual priors to compute the rollout rewards.

**Compliance With Llm Reviewing Policy:**

Affirmed.

**Final Justification:**

The rebuttal clarified the authors’ intention, but I remain unconvinced that MCTS itself is the essential contribution. Overall, I appreciate the clarification and am willing to raise my score to 3 (weak reject).

**Key Questions For Authors:**

1.How were the default non-zero weights (e.g., lamda_hot, lamda_obj) determined for this zero-shot, training-free framework?

2.What would be the performance if RefChess uses CLIP+SAM as HybridGL does?

**Limitations:**

yes

**Strengths And Weaknesses:**

## Strengths:

1. The paper reformulates proposal selection in RIS as a decision-making problem.
2. The paper designs a unified, constraint-aware proposal score that aggregates heterogeneous evidence.

## Weaknesses:

1. Misalignment between Motivation and MCTS Application.

   The application of MCTS appears overclaimed for the problem being solved. MCTS is designed for sequential decision-making tasks where early actions directly influence future states and available choices. However, in this paper, the "chain" of proposals is essentially an unordered subset. The order in which the distractor proposals are sampled and appended does not fundamentally influence the final reward. Consequently, framing this as a "game tree" traversal may be an over-formulation.
2. Theoretical Ambiguity in the Stability-Aware Reward

   The paper claims distractors create "randomized contexts" to measure a primary proposal's stability. However, Eq. 7 merely adds the mean and subtracts the standard deviation of **uniformly sampled, independent** distractor scores. The authors should further clarify how this independent formulation meaningfully captures "contextual stability." For example, how can a true mask with a lower initial score outcompete the incorrect ones with higher initial scores?
3. Flawed Ablation Study Design

   The empirical validation of the MCTS module in Table 2 lacks a fair baseline comparison. The text indicates that the "Greedy" baseline directly selects the proposal with the **highest CLIP score** . In contrast, the MCTS method relies on a much more comprehensive and robust scoring function **Si (eq5)**. To rigorously prove the efficacy of MCTS, the authors should provide a fair comparison.

---

> ### Author Rebuttal · Authors · 2026-03-30
>
> We thank the reviewer for the helpful comments and address the concerns regarding the MCTS formulation, Eq. (7), the ablation study, hyperparameters, and comparison with HybridGL below.
>
> 1. On the Motivation and Application of MCTS
>
> We agree that RIS is not a classical sequential control problem. Our use of MCTS is not motivated by temporal dynamics, but by the need to perform robust proposal selection through searching over distractor subsets under a limited computational budget. In our formulation, the “chain” is one sampled contextual subset, where the first element is the primary candidate and the remaining elements are distractors. The reward depends on how the candidate behaves under that distractor subset, rather than on the order in which distractors are added. Since exhaustively enumerating all distractor subsets is computationally expensive, we use MCTS as a budgeted combinatorial search procedure to efficiently explore this space.
>
> We also agree that describing this process as a “game tree” may have been too strong. Inspired by chess, we instead reformulate RIS as a search problem over contextual perturbations, so the resulting tree is better viewed as a search tree for robust proposal selection rather than a sequential decision process.
>
> 2. On Eq. (7) and Contextual Stability
>
> The "Stability-Aware Reward" in Eq. (7) is intended as an operational surrogate for proposal robustness under contextual perturbations. Specifically, it combines three aspects: (i) the intrinsic score of the primary proposal, which reflects its standalone compatibility with the expression; (ii) the average score against distractors ($\mu$), which measures whether the proposal remains competitive when different distractors are introduced; and (iii) a variance penalty ($\sigma$), which discourages proposals whose scores fluctuate strongly across distractor subsets. In this sense, “stability” refers not to the highest isolated score, but to consistently strong performance under different contextual perturbations. Accordingly, Eq. (7) favors proposals that remain strong on average and less sensitive across distractor subsets.
>
> This intuition is illustrated in Fig. 2 (use example image as a test) for the query “man wears a red coat.” Under greedy ranking, the distractor (“blue coat man”) scores higher than the target (“red coat man”) (14.61 vs. 13.38) and is selected. After MCTS re-ranking, however, the target rises to 17.38, surpassing the distractor’s 15.15, indicating greater stability under distractor contexts. We will clarify this interpretation of Eq. (7) more explicitly in the revision.
>
> 3. On the Fairness of the Ablation Study
>
> We apologize for the unclear wording. Both the Greedy baseline and MCTS use the same unified scoring function ($S_i$ in Eq. 5), which integrates language decomposition, object-centric cues, and spatial heatmap evidence. Greedy therefore does not use a naive CLIP score; Table 2 isolates the effect of the selection strategy on top of the same score. Therefore, the ablation is designed to isolate the effect of contextual search (Greedy vs. MCTS), rather than changes in the scoring function itself. We will revise the text to state that the same unified scoring function $S_i$ is used for both selection strategies.
>
> 4. On Hyperparameter Determination
>
> The default non-zero weights were chosen based on the relative scales of the score components, rather than by arbitrary tuning. In our formulation, the CLIP similarity term (Eq. 2) is typically on the order of $10^1$, whereas the spatial heatmap terms ($E_{hot}, E_{obj}$) are on the order of $10^{-1}$. We therefore set $\lambda_{hot}$ and $\lambda_{obj}$ to 10 so that these terms contribute on comparable numerical scales.
>
> We also evaluated the sensitivity of this choice on the RefCOCO val set. When $\lambda_{hot},\lambda_{obj}$ was reduced from 10 to 1.0, the mIoU decreased from 55.19 to 54.12; when further reduced to 0.1, the mIoU became 53.96, which is close to the zero-weight setting reported in Table 4. These results show that the default setting is determined by scale matching across heterogeneous terms. In addition, although the parameter choice does affect performance, the impact is moderate rather than implying strong dependence on a single fragile setting. We will include a fuller summary of this sensitivity analysis in the revision.
>
> 5. On Comparison with HybridGL (CLIP+SAM)
>
> To control for backbone differences in the comparison with HybridGL, we additionally adapted our method to the same CLIP+SAM backbone used by HybridGL. Under this backbone-controlled setting on the RefCOCO val set, our method achieves 51.03 mIoU and 49.74 oIoU. These results compare favorably with HybridGL’s reported performance (49.48 mIoU and 41.81 oIoU), suggesting that the gain of RefChess is not solely due to backbone differences, but is retained even when using the same backbone.

---

> > ### Author Rebuttal · Reviewer_ynba · 2026-04-03
> >
> > I thank the authors for their responses. In fact, they also acknowledge that RIS is “not a classical sequential control problem” and that describing it as a “game tree” may have been too strong. This is exactly my main concern. Given that, I still feel the title’s emphasis on Monte-Carlo Move Selection is somewhat overstated.
> >
> > Overall, I appreciate the clarification and am willing to raise my score to 3 (weak reject).

---

> > > ### Author Response · Authors · 2026-04-03
> > >
> > > Thank you again for the thoughtful follow-up. We fully understand your concern that RIS is not a classical sequential control problem, and we agree that our earlier wording may have overstated the role of MCTS by describing the process too strongly as a "game tree" or "move selection" problem.
> > >
> > > To clarify, our intention is not to claim that RIS is inherently a canonical MCTS problem, nor that MCTS is the only valid formulation. Rather, the core contribution of RefChess is a training-free framework for robust proposal selection under contextual perturbations, where distractor subsets are used to evaluate the consistency of candidate masks. Within this framework, MCTS serves as a practical budgeted search strategy for efficiently exploring the combinatorial space of distractor subsets.
> > >
> > > Under this more conservative framing, we believe the work remains technically meaningful and empirically supported. In particular, we also clarified during rebuttal that:
> > >
> > > (1) Eq. (7) is intended as an operational surrogate for robustness under contextual perturbations, combining intrinsic proposal quality, average competitiveness across distractor subsets, and variance penalization;
> > >
> > > (2) the Greedy and MCTS ablations are fair because both use the same unified scoring function, isolating the effect of the search strategy rather than changing the scoring function;
> > >
> > > (3) the non-zero weights are chosen according to scale matching across heterogeneous terms, and the sensitivity analysis shows moderate rather than fragile dependence;
> > >
> > > (4) the gain is not solely due to backbone choice, as our backbone-controlled comparison using the same CLIP+SAM setup as HybridGL still shows favorable performance.
> > >
> > > Importantly, under the controlled ablation where Greedy and MCTS use the same unified scoring function and differ only in the selection strategy, replacing Greedy with MCTS still yields clear gains: on RefCOCO, performance improves from 40.77/29.95 to 55.19/48.47 (mIoU/oIoU), and on RefCOCOg from 46.37/34.49 to 50.63/42.85. We view this as evidence that, even under the more conservative framing, the search component provides meaningful empirical benefit beyond one-step ranking.
> > >
> > > To directly address your concern, we are happy to revise the title, abstract, and method description to substantially soften the emphasis on "Monte-Carlo Move Selection." For example, we would revise the title to: RefChess: Training-Free Contextual Search for Zero-Shot Referring Image Segmentation, which more accurately presents the method as a contextual-search-based framework for robust proposal selection rather than a strict sequential-control formulation.
> > >
> > > Under this revised framing, we hope the contribution may be viewed primarily as introducing a simple yet effective contextual search paradigm for training-free RIS, with MCTS as one effective mechanism for realizing that search under limited computation. We sincerely appreciate your comments, which we believe will help us present the work more precisely and conservatively.

---

### Official Review · Reviewer_8YuJ · 2026-03-12

**Soundness:** 2
**Presentation:** 3
**Significance:** 3
**Originality:** 2
**Overall Recommendation:** 5
**Confidence:** 3

**Summary:**

The paper proposes RefChess, a training-free framework for zero-shot referring image segmentation. Instead of ranking segmentation proposals independently, it models proposal selection as a decision-making process and applies Monte Carlo Tree Search (MCTS) to evaluate competing regions. By combining language cues, vision–language similarity, and spatial information, the method selects more reliable segmentation results. Experiments show it improves robustness and accuracy over existing proposal-ranking approaches.

**Compliance With Llm Reviewing Policy:**

Affirmed.

**Final Justification:**

The responses address my concerns and some misunderstandings, so I'm happy to increase my score.

**Key Questions For Authors:**

Please see the previous section.

**Limitations:**

Please see the previous section.

**Strengths And Weaknesses:**

Strengths:
1. Formulate proposal selection in RIS as a decision-making problem and using Monte-Carlo Tree Search to solve it is a novel idea.
2. Good results are achieved for referring image segmentation.
3. The paper is generally easy to understand.

Weaknesses:

1. The motivation for using Monte Carlo Tree Search is not sufficiently clear. For example, the paper should better explain what advantages MCTS provides and which specific limitations of previous methods it is intended to address.
2. In Table 2, why the proposed method does not show a clear advantage in oIoU, and even performs worse on some datasets?
3. In Table 1, most of the compared training-free methods are relatively outdated. Only one method was published in 2025, while the others were published before 2023.
4. Could the authors clarify the specific innovations related to the use of Monte Carlo Tree Search in this work? For example, is the method based on a standard MCTS framework, or are there task-specific modifications designed for the RIS problem?

---

> ### Author Rebuttal · Authors · 2026-03-30
>
> We sincerely thank the reviewer for the insightful comments and the effort invested in reviewing our work. We also appreciate the opportunity to clarify the motivations, performance, and contributions of our work. Below, we address each of the reviewer's concerns point-by-point:
>
> 1. On the Motivation for MCTS
>
> We thank the reviewer for this question. The core motivation lies in addressing the brittleness of isolated proposal ranking. Traditional zero-shot RIS methods score proposals largely independently, ignoring the competitive context. This often leads to failure when multiple regions partially satisfy the expression.
>
> RefChess instead evaluates whether a candidate remains favorable when competing proposals are introduced as distractors. Since exhaustively evaluating all possible distractor combinations is intractable, we use MCTS as a practical budgeted search mechanism for exploring the space.
>
> 2. On oIoU Performance (Interpretation of Table 2)
>
> We respectfully clarify the role of Table 2. Table 2 is an ablation designed to isolate the effect of the selection strategy (MCTS vs. Greedy) under our framework, rather than the main comparison against prior methods. The primary comparison with zero-shot baselines is given in Table 1. As shown in Table 1, our method achieves the strongest oIoU among zero-shot baselines across different datasets.
>
> The less favorable mIoU behavior on RefCOCOg reflects a known limitation of proposal-based methods in cluttered scenes, as discussed in Appendix A.3. However, our oIoU remains consistently strong, indicating that the method still provides robust instance-level selection even when fine-grained mask quality is more challenging in cluttered settings.
>
> 3. On Comparison with State-of-the-Art (Including 2025 Works)
>
> Table 1 already includes several recent zero-shot RIS methods, such as Pseudo-RIS (2024), VLM-VG (2024), CaR (2024), and HybridGL (2025). In response to the reviewer’s concern, we further examined the most recent related work and additionally included LGD (Li et al., 2025) in our comparison. Although LGD appeared close to our submission deadline, we compared against it under the same evaluation setting. Under this comparison, RefChess achieves stronger results than LGD (e.g., 55.19 mIoU vs. 45.41 on RefCOCO val; 48.63 mIoU vs. 47.23 on RefCOCO+ val). These comparisons suggest that RefChess remains competitive with very recent zero-shot RIS methods. We will clarify this additional comparison in the revision.
>
> 4. On Innovations in MCTS
>
> To address the brittleness of isolated proposal ranking, we propose RefChess, which evaluates whether a candidate remains favorable when competing proposals are introduced as distractors. Since exhaustively evaluating distractor subsets is combinatorial and intractable, we use MCTS as a practical budgeted search mechanism for this space.
>
> The method follows the MCTS framework, but the search formulation is specifically designed for RIS. Specifically, we define a search process in which the state corresponds to a candidate proposal under sampled distractor context, the action adds an unused proposal as an additional distractor, and the reward evaluates robustness under contextual perturbation. Under this formulation, MCTS is not used as a generic plug-in, but as a task-specific search procedure that replaces isolated greedy ranking with context-aware robust selection in RIS. We will clarify this positioning more explicitly in the revision.
>
> References: Li, J., Xie, Q., Gu, R., Xu, J., Liu, Y., and Yu, X. Lgd: Leveraging generative descriptions for zero-shot referring image segmentation. 2025.

---

> > ### Author Rebuttal · Reviewer_8YuJ · 2026-03-31
> >
> > Thank authors for the responses that address my concerns and some misunderstanding. I'm happy to increase my score.

---

> > > ### Author Response · Authors · 2026-04-01
> > >
> > > Thank you for your thoughtful follow-up; we are very glad that our response helped clarify the misunderstandings and address your concerns.

---

### Official Review · Reviewer_5ern · 2026-03-12

**Soundness:** 3
**Presentation:** 3
**Significance:** 3
**Originality:** 3
**Overall Recommendation:** 4
**Confidence:** 4

**Summary:**

This paper works on zero-shot referring image segmentation (RIS), and tries to prevent from the failure of existing SAM+CLIP methods to account for contextual interactions among visually similar segmentation proposals. Specifically, authors propose RefChess, a training-free framework that reformulates proposal selection as a decision-making problem via Monte-Carlo Tree Search (MCTS). RefChess models each SAM-generated proposal as a "chess move" and evaluates its robustness under simulated distractor perturbations using a stability-aware reward, which integrates language decomposition (subject, attribute, relation parsing), CLIP-based semantic compatibility, object-centric cues, and spatial guidance (location, size heatmaps). Extensive experiments have been conducted to show the effectiveness of each module design.

**Compliance With Llm Reviewing Policy:**

Affirmed.

**Final Justification:**

Most of my concerns have been well addressed. Although few concerns cannot be resolved in the current stage, they are not crucial, so I keep my rating as weak accept.

**Key Questions For Authors:**

See weaknesses.

**Limitations:**

yes

**Strengths And Weaknesses:**

**Strengths**:

1. The chess-inspired MCTS framework introduces a novel way to model contextual perturbations, filling a gap in existing zero-shot RIS methods that ignore inter-proposal interactions.
2. The integration of linguistic structure, semantic similarity, and spatial cues provides a more comprehensive evaluation of proposals than single-scale CLIP scoring.
3. The paper effectively justifies the need for robustness evaluation, with Figure 1 and Section 1 highlighting the failures of greedy CLIP-based ranking in complex scenes.

**Weaknesses**:

1. The paper does not explore the impact of some hyperparameters (e.g., rollout depth​), leaving uncertainty about the framework's sensitivity to parameter tuning.
2. The lightweight language parser's accuracy is not validated across diverse referring expression types (e.g., highly relational or ambiguous phrases), raising questions about its generalizability.
3. The MCTS simulation process may introduce significant latency, but no efficiency analysis is provided.
4. Please include some failure cases for discussion.
5. The examples in Figure 3 are instance segmentation, I'm interested if the model can handle semantic segmentation as well, e.g., segment all cars in the first row.

---

> ### Author Rebuttal · Authors · 2026-03-30
>
> We thank the reviewer for the succinct summary and for recognizing the novelty of our "chess-inspired" MCTS framework in modeling contextual perturbations. We also appreciate the constructive feedback regarding hyperparameter sensitivity, parser generalizability, efficiency, and the scope of the task. We realize that some of these points were not sufficiently highlighted in the main text, even though relevant analyses are included in the appendix. Below, we address each point with specific references to our analysis.
>
> 1. Regarding Hyperparameter Sensitivity (Rollout Depth & Budget)
>
> We analyzed hyperparameter sensitivity in Table 2 and Appendix A.2.
> For rollout depth, varying (D) from 5 to 15 leads only to small performance fluctuations, indicating the method is not brittle to this choice. Even a shallow depth yields clear gains over the Greedy baseline.
> We also observe that reducing the simulation budget to 256 still achieves consistent improvements over Greedy across all three datasets (RefCOCO, RefCOCO+, RefCOCOg).
> In addition, Appendix A.2 further shows that the reward coefficients ($\lambda,\gamma$) exhibit stable behavior within small-to-moderate ranges, suggesting that the framework is not overly sensitive to moderate hyperparameter variation.
>
> 2. Regarding Lightweight Parser Generalizability
>
> Our parser is designed not as a standalone solver,but as a lightweight module that exposes interpretable structure (subject, attribute, relation, etc.) to guide the reward function. We evaluate it across diverse expression styles, including short spatial expressions (RefCOCO), attribute-centric phrases without location words (RefCOCO+), and longer relational descriptions (RefCOCOg). As shown in Table 1, the consistent performance gains across these datasets suggest that the parser generalizes beyond a single narrow expression type.
>
> Crucially, the scoring function also retains a global matching term ($S(mi∣T)$), so the method does not rely solely on local parsing. This helps stabilize cases where parsing is incomplete or ambiguous, and makes the parser a soft decomposition signal rather than a brittle hard dependency. In addition, Table 4 provides direct evidence of the parser's contribution. Removing specific parsed components (subject/attribute/relation) leads to performance drops. Together, these results suggest that the parser contributes meaningfully within the overall scoring framework while remaining robust across different expression styles.
>
> 3. Regarding Efficiency and Latency
>
> We agree that this aspect should have been surfaced more clearly in the main text. Efficiency is analyzed in Appendix A.1 and Table 6 through an explicit accuracy–latency trade-off study.
>
> On a single RTX 4090, RefChess runs at 1231.21 ms, which is slower than HybridGL (862.43 ms) but faster than TAS (2593.87 ms). Importantly, reducing the simulation budget to 256 steps lowers runtime to 842.34 ms, which is comparable to HybridGL, while achieving 53.84 mIoU on RefCOCO val. If heatmap guidance is further removed, our method achieves an even lower runtime of 663.31 ms while accuracy still compares favorably with HybridGL. This suggests that RefChess can be adjusted to different deployment requirements depending on whether speed or accuracy is prioritized. We will clarify this practical trade-off more explicitly in the revision.
>
> 4. Regarding Failure Cases
>
> We analyzed failure cases in Appendix A.3 and visualize representative examples in Figure 5. The main failure modes fall into three categories: (1) insufficient coverage or low quality of SAM-generated proposals; (2) expressions requiring implicit or highly specific attribute reasoning; and (3) cluttered scenes with multiple plausible candidates where relational cues remain insufficient. These cases suggest that the current bottlenecks are mainly proposal quality and fine-grained/implicit reasoning, which we will discuss more explicitly as limitations in the revision.
>
> 5. Regarding the Scope: Instance vs. Semantic Segmentation
>
> We appreciate the reviewer’s question regarding semantic segmentation (e.g., "segment all cars in the first row"). Our work targets Zero-Shot Referring Image Segmentation (RIS), where the goal is to identify a single specific instance based on a referring expression. Accordingly, our MCTS framework is designed to select a single target proposal ($i_{0}$) that best matches the unique target described. This design is consistent with the benchmarks we used (the RefCOCO series), where each expression is paired with a single-instance ground-truth mask.
>
> We agree that extending RefChess to set-valued queries (semantic segmentation) or targetless rejection is both valuable and challenging, but it goes beyond the scope of the current paper. We will make this scope and limitation more explicit in the revision and highlight it as a direction for future work.

---

> > ### Author Rebuttal · Reviewer_5ern · 2026-04-02
> >
> > Most of my concerns have been well addressed. Although few concerns cannot be resolved in the current stage, they are not crucial, so I keep my rating as weak accept.

---

> > > ### Author Response · Authors · 2026-04-03
> > >
> > > Thank you for your positive feedback and for recognizing that most of your concerns have been well addressed. We also appreciate your understanding regarding the few remaining issues, and we will further polish the final version to improve the presentation and clarity where possible.

---

### Official Review · Reviewer_KEf6 · 2026-03-13

**Soundness:** 3
**Presentation:** 3
**Significance:** 3
**Originality:** 3
**Overall Recommendation:** 4
**Confidence:** 4

**Summary:**

This paper studies training-free zero-shot referring image segmentation. Instead of selecting SAM proposals via one-step ranking, it reformulates proposal selection as a robust decision-making problem under contextual perturbations. The proposed method, RefChess, combines CLIP-based proposal scoring, language decomposition, object/spatial priors, and Monte Carlo Tree Search to select proposals that remain stable under distractor contexts. Experiments on RefCOCO, RefCOCO+, and RefCOCOg show substantial gains over prior zero-shot baselines, especially on RefCOCO and RefCOCO+.

**Compliance With Llm Reviewing Policy:**

Affirmed.

**Final Justification:**

Thank you for the clarification. It addresses some of my concerns, but not sufficiently, so I maintain my original score.

**Key Questions For Authors:**

Please refer to the Weaknesses section.

**Limitations:**

yes

**Strengths And Weaknesses:**

Strengths.
The paper has a clear and well-motivated problem statement: proposal selection in zero-shot RIS is brittle when visually similar distractors satisfy partial parts of the referring expression. Reformulating proposal selection as robust search rather than independent scoring is intuitive and practically meaningful. The method is training-free, easy to implement on top of existing foundation models, and empirically strong. The greedy-vs-MCTS ablation is particularly convincing, showing that the search component matters.

Weaknesses.
This article is more like a well-designed reasoning system rather than a new learning method. What remains unclear is whether it is the contextual reasoning of MCTS that is effective, or stronger scoring combined with heatmap/object priors. Particularly, since the language term weight is much smaller than the object/heatmap term, the narrative of 'language-driven robust decision-making' is not as cleanly supported by experiments. Another practical issue is that the additional latency introduced by MCTS is significant.

---

> ### Author Rebuttal · Authors · 2026-03-30
>
> Thank you for your thoughtful review and for recognizing the clear problem statement and empirical strength of RefChess. We appreciate your insights regarding the nature of our contribution, the role of MCTS, the weighting of terms, and the latency considerations. Below we address your specific concerns:
>
> 1. On the Nature of Contribution (Reasoning System vs. Learning Method):
>
> We agree that RefChess is better characterized as a training-free decision framework rather than a new trainable backbone. Our core contribution lies in reformulating proposal selection in zero-shot RIS from independent single-step ranking into robust selection process under contextual perturbations. In conventional scoring-based methods, proposals are evaluated largely in isolation, which can be brittle when multiple regions partially satisfy the expression. RefChess instead evaluates whether a candidate remains favorable when competing distractors are introduced, with MCTS providing a practical mechanism for searching this contextual space.
>
> 2. On the Effectiveness of Contextual Reasoning (MCTS) vs. Priors:
>
> To isolate the effect of contextual reasoning, both Greedy and MCTS use the same scoring function with heatmap/object priors; the only difference is the selection strategy. In our MCTS formulation, the reward function explicitly evaluates the stability of a candidate proposal when competing regions (distractors) are introduced as context. As shown in Table 2 of the main paper, replacing Greedy with MCTS under this controlled setting yields substantial improvements: on RefCOCO, mIoU/oIoU increases from 40.77/29.95 to 55.19/48.47; and on RefCOCOg, from 46.37/34.49 to 50.63/42.85. This suggests that the improvement does not come simply from stronger priors or a different scoring function, but from evaluating candidate stability under distractor perturbations.
>
> We also observe  from Appendix (A.1 Table6) that reducing the simulation budget to 256 steps still yields 53.84/47.02 on RefCOCO, demonstrating that the gain is retained even under a substantially smaller search budget.
>
> 3. On Language Term Weights and "Language-Driven" Narrative:
>
> We agree that coefficient magnitudes can appear counterintuitive at first glance, but they are not directly comparable across terms because the components in Eq. (5) operate on different numerical scales. As defined in Section 3.3, language terms are CLIP-scaled cosine similarities (involving $e^{τ}$), whereas $E_{obj}$ and $E_{hot}$ are spatial averages over proposal regions;  in practice, these are typically on the order of $10^{1}$ and $10^{-1}$, respectively. The coefficients in Eq.(5) therefore serve primarily to align numerical scales rather than to reflect relative importance.
>
> More importantly,  the strongest evidence comes from our ablations in Table 4 of the main paper, which show that the language terms are indispensable to performance. Removing the full-expression matching term causes clear mIoU drops on all three datasets (RefCOCO: 55.19 → 51.72; RefCOCO+: 48.63 → 45.26; RefCOCOg: 50.63 → 46.98). Similarly, removing the subject term reduces RefCOCO to 52.76. These results are the clearest indication that the primary discriminative signal comes from the language terms.
>
> By contrast, reducing the object/heatmap weights leads to smaller performance changes. On the RefCOCO validation set, when both $\lambda_{hot}$ and $\lambda_{obj}$ are reduced from the default value of 10 to 1 and 0.1, the mIoU decreases to 54.12 and 53.96, respectively. This suggests that the object/heatmap cues are useful, but their effect is weaker than the much larger drops observed when key language terms are removed. In particular, the result at 0.1 is already close to the zero-weight case reported in Table 4, which is consistent with the view that the object/heatmap terms provide supplementary support rather than the dominant discriminative signal.
>
> Taken together, these results support a language-driven interpretation of RefChess: language provides the primary semantic discrimination, while the heatmap/object cues act as complementary stabilizers, especially in the presence of similar distractors. We will clarify this scale-matching interpretation and include a broader sensitivity summary in the revision.
>
> 4. On Latency and Practicality:
>
> As shown in Appendix A.1, our method runs at 1231.21 ms, which is slower than HybridGL (862.43 ms) but faster than TAS (2593.87 ms). Importantly, reducing the simulation budget to 256 steps lowers runtime to 842.34 ms, which is comparable to HybridGL, while achieving 53.84 mIoU on RefCOCO val. If heatmap guidance is further removed, our method achieves an even lower runtime of 663.31 ms while accuracy still compares favorably with HybridGL. This suggests that RefChess can be adjusted to different deployment requirements depending on whether speed or accuracy is prioritized. We will clarify this practical trade-off more explicitly in the revision.

---

> > ### Author Rebuttal · Reviewer_KEf6 · 2026-04-02
> >
> > While the rebuttal partially alleviates my concerns, I still find that some of the paper’s central claims remain stated more strongly than the current evidence warrants, so I would encourage more careful wording in the final version and keep my original score unchanged.

---

> > > ### Author Response · Authors · 2026-04-03
> > >
> > > Thank you for your thoughtful follow-up and for acknowledging that our rebuttal has partially addressed your concerns. We appreciate your suggestion, and in the final version we will revise the wording carefully to ensure that our central claims are stated with appropriate caution and fully supported by the current evidence.

---

### Decision · Program_Chairs · 2026-04-30

**Decision:**

Accept (regular)

**Comment:**

This paper proposes a training-free contextual search framework based on MCTS for zero-shot RIS and shows strong empirical results on several standard benchmarks. Reviewers generally found the method practically meaningful and the experimental results solid, and most concerns were largely addressed during the rebuttal.

The main remaining issue concerns the framing of the method, especially the role of MCTS. One reviewer raised a remaining concern about whether RIS naturally fits a sequential decision-making formulation, while another noted that some of the main claims could be stated more carefully. In the rebuttal, the authors clarified that the core idea is proposal selection through contextual search over distractor subsets, with MCTS serving as a practical search mechanism within this framework. The authors also indicated that they are willing to revise the title, abstract, and method description to substantially soften the main claim.

After the rebuttal and subsequent discussion, the remaining concern appears to be more about framing and positioning than about technical soundness or empirical validity. Therefore, the AC recommends acceptance. If accepted, the final version should revise the title and some of the main claims, and present the method more precisely as a contextual search framework, so that the presentation better matches both the current empirical support and the actual scope of the technical contribution.